# Differences in the acceptance of wife abuse among ethnic minority Garo and Santal and mainstream Bengali communities in rural Bangladesh

**Rabiul Karim**[1,2]☯*, **Tanzima Zohra Habib**[1‡], **Sadequl Arefin**[1‡], **Hafijur Rahman**[1‡], **Suchona Rahman**[1‡], **Katarina Swahnberg**[2☯]

**1** Department of Social Work, University of Rajshahi, Rajshahi, Bangladesh, **2** Department of Health and Caring Sciences, Faculty of Health and Life Sciences, Linnaeus University, Kalmar, Sweden

☯ These authors contributed equally to this work.
‡ These authors also contributed equally to this work.
* Rabiul.Karim@lnu.se

**Data Availability Statement:** All relevant data are within the manuscript and its Supporting Information files.

## Abstract

Studies on wife abuse in Bangladesh predominantly include the mainstream *Bengali* population, although there are at least 27 ethnic minority communities including a few 'female-centered' matrilineal groups living in the country. This study explored ethnic differences in the attitudinal acceptance of wife abuse among matrilineal ethnic minority *Garo*, patrilineal ethnic minority *Santal*, and mainstream patriarchal *Bengali* communities in rural Bangladesh. Adopting a cross-sectional design, the study included 1,929 women and men randomly selected from 24 *Garo*, *Santal*, and *Bengali* villages. Multivariate Poisson regression was performed to predict the number of contextual events, where the respondents attitudinally endorsed wife abuse. Of the sample, 33.2% were from *Garo*, 33.2% from *Santal*, and 33.6% from the *Bengali* communities. The acceptance of wife abuse was high in the sample; specifically, 34.1% of the respondents accepted physical wife abuse, 67.5% accepted emotional abuse, and 71.6% accepted any abuse (either physical or emotional) at least on one contextual reason provided in a 10-item scale. The mean for accepting any abuse was 3.0 ($SD = 2.8$), emotional abuse 2.3 ($SD = 2.2$), and physical abuse 0.8 ($SD = 1.4$). The study showed that the rates of accepting any abuse and physical abuse were respectively 16% and 56% lower among *Garo* as well as 14% and 33% lower among *Santal* than that of the *Bengali* community. Data also revealed that individual level factors like younger age, higher education, prestigious occupation as well as family level factors such as higher income, female mobility, and female family authority were inversely associated with the acceptance of wife abuse in the sample. It appears that the gender regime of a society has a great influence on the attitudes toward wife abuse. We argue that a comprehensive socio-cultural transformation of the patriarchal societies into a gender equal order is imperative for the prevention of widespread wife abuse in the country.

**Funding:** The authors acknowledge partial financial supports from Rajshahi University, Swedish Research Council and Linnaeus University. Arefin and Rabiul received partial funding from Rajshahi University. This fund was used for piloting the study including instrument development and validation. Rabiul received a fieldwork grant from Swedish Research Council as well as Katarina received partial funding from Linnaeus University. These funds were used for this survey. There was no additional external funding received for this study. The funders had no role in study design, data collection and analysis, decision to publish, or preparation of the manuscript.

**Competing interests:** The authors have declared that no competing interests exist.

# Introduction

Acceptance of wife abuse (AWA) is a critical factor of the intimate partner violence extensively prevalent in many South Asian countries [1–4]. Studies conducted in African and Middle-eastern countries such as Ethiopia, Israel, and Nigeria also demonstrated that people's attitudinal justification of wife abuse is a significant feature of the intimate partner violence against women [1, 5–8]. AWA is also a widely talked about topic in the discussion on the primary prevention of violence against women worldwide. In order to prevent violence against women, scholars and practitioners often advocate community-based interventions that will reduce AWA among people [9–11]. This is because many people in patriarchal societies believe that a husband has a right to 'punish' his wife [3, 12]. Societies support or tolerate a husband's abusive behaviors toward a wife under certain circumstances, e.g., if the wife argues with her husband. This social acceptance of wife abuse is extensively obvious in Bangladesh. It is assumed that husbands abuse their wives as a means of maintaining their family authority over women [13], whereas the abusive act is often accepted by other family members, relatives, and neighbors [14, 15]. The concept 'acceptance of wife abuse' may refer to the context-specific reasons where people rationalize 'the privilege of a husband' to punish/abuse his wife [12].

A previous study revealed that almost 80% of the married women in rural Bangladesh justified that a husband has a right to beat his wife [12]. This is quite similar to the actual extent of wife abuse in the country, where nearly eight in ten women experience physical or sexual abuse from their husbands [16]. There were also examples of wife beating where the abuse was justified in relation to women's failure to fulfil expected family duties [17–19]. There were also instances where the abuse was used as a socially accepted means of controlling one's wife [15].

Justifying husbands' abusive behavior toward one's wife as a 'normal conduct' is a common attitudinal character among both perpetrators and victims. There is a trend among abusive husbands, wherein they try to justify the violence by indicating 'women's faults' [15]. This is how wife abuse is socially justified and treated as 'normal.' Many female victims also try to accept this as 'normal.' Since accepting wife abuse as a 'normal' behavior has many negative consequences on women and children, an understanding of AWA in different ethno-cultural contexts might provide guidance in preventing the socio-cultural problem in an effective way.

The culture of wife abuse is deeply rooted within the traditions of a patriarchal society. According to a proposition of the patriarchy theory, wife abuse is a symptom of a patriarchal society reflected in male authority, leading to the 'rights' of a husband to control one's wife [20–22]. Therefore, we assumed that AWA might not be obvious in the ethnic minority communities where gender equality is relatively high or where women enjoy independence to a greater extent.

A previous cross-cultural study conducted with ethnic groups in Africa revealed that different cultural factors such as female mobility and participation in production, marriage patterns, and living arrangements may influence AWA [23]. The study showed that the ethnic groups practiced endogamy and patrilocality; moreover, in the groups where women participated less in the 'productive activities,' they had more acceptance of the violence against women [23]. The study also considered the economic value of women because of bride-price. It was revealed that where men attributed a greater value to women, there seemed to be a lower acceptance of wife abuse [23], although the value of women could also be understood by focusing on the role/status of women in female-centered or matrilineal ethnic communities. Studies based in Tanzania and Bangladesh also discussed the influence of societal and community level factors on AWA. However, these studies generally lack cultural as well as ethnic dimensions of the attitudes toward wife abuse [24, 25]. Another multi-country study based on the Demographic and Health Survey (DHS) data from 49 low- and middle-income countries

identified the influence of societal level factors on AWA. They found that a lower socio-economic status of the women in a society appeared to be inversely associated with the acceptance of wife abuse among both women and men in those societies [26].

Using a similar type of DHS instrument, previous studies also measured the attitudes toward wife beating in other countries like Armenia, Egypt, Ethiopia, India, Nepal, Nigeria, Pakistan, Palestine, etc. [e.g., 1, 3, 5, 8, 27, 28]. These studies were mostly confined to the acceptance of wife beating. However, a comprehensive study should include the acceptance of both physical abuse and emotional abuse. We assumed that the magnitude of the acceptance of a wife's emotional abuse could be different from that of the physical abuse.

Previous studies provided little empirical evidence on elucidating how socio-cultural differences may influence the AWA in rural Bangladesh, although this could be well-understood by focusing on the ethnic minority communities (including the matrilineal groups). Previous studies also generally failed to focus on the acceptance of the wife's emotional abuse. In general, there is a lack of studies on wife abuse among the ethnic minorities in Bangladesh, although there are at least 27 ethnic minority communities living in different parts of the country. Previous studies only focused on wife-beating as well as predominantly included the majority *Bengali* population.

Bangladesh is a lower-middle income country situated in South Asia, where roughly 24.3% of the people live below the poverty line–earning less than two USD a day [29]. There are almost 162 million people. The mainstream *Bengali* ethno-linguistic community is traditionally patriarchal, consisting of 98% of the total inhabitants. The population includes Muslims 89.0%, Hindus 10.7%, Buddhists 0.6%, and Christians 0.4%, respectively. On the other hand, the Government of Bangladesh recognized a total of 27 small ethnic groups (comprising 2% of the population), although different rights-based organizations have claimed that there may be more than 45 ethnic minority communities living in different parts of the country. Most of the ethnic minorities e.g., Chakma, Marma, Manipuri, Santal, etc. are patriarchal; there are also 'female-centered' matrilineal societies such as Garo and Khasi. All these ethnic minorities have their own language, norms, customs, inheritance rules for properties, marriage system, gender order, etc. The current study is an attempt to examine the differences in the attitudinal acceptance of wife abuse among 'female-centered' ethnic minority *Garo*, somewhat male-dominated ethnic minority *Santal*, and typical patriarchal mainstream *Bengali* communities in rural Bangladesh.

## The Bengali ethnicity

The mainstream Bengali ethnic community is an example of a classical patriarchal society. In general, women are deprived of many human rights, particularly in the rural *Bengali* community. Gender inequality is very high in Bangladesh. The country ranks 129 out of 162 countries in the Gender Inequality Index (GII) [30]. The high gender inequality in Bangladesh is mostly reflected by its mainstream population. Gender inequality among the *Bengali* community in Bangladesh is characterized by veiled seclusion among women (restricted mobility); women's limited access to economic resources, employment and higher education; male guardianship and control over women's life choices; male dominated social institutions; and a high prevalence of violence against women [12, 16]. In rural Bangladesh, many men use wife abuse as a part of their socially accepted patriarchal 'right' to control the wives [12, 18]. The problem of men's wife-abusive behaviors in the society is probably deeply rooted in the structure of its patriarchal family institutions. In rural Bengali community, women are considered as the non-productive members of the household, while men are regarded as the economic providers. Therefore, sons/men inherit most of their parental properties; women move to their husband's

home (patrilocality) after marriage and children adopt the patrilineal descent. Married women are expected to: stay at home, perform all the home-making chores; and take care of the husband, children, and in-laws. People also believe that the husband should be the guardian/head of the family, and that the wife should obey her husband [31]. However, the ethnic minority communities in Bangladesh, e.g., *Garo* and *Santal*, are socio-culturally different from the mainstream *Bengali* community.

## The Garo ethnicity

The matrilineal *Garo* community is one of the largest ethnic groups (about 200,000 in total), living in the north-eastern hilly parts of Bangladesh. *Garo* people migrated from a north-western Chinese province to the forested areas of the *Garo* Hills region nearly 4,000 years ago. They still continue to live in the same region across the border in northern Bangladesh and the adjacent Meghalaya state of India. Historically, the *Garo* community maintains a different socio-cultural tradition from the *Bengali* community. *Garos* are sometimes described as a matriarchal society (like *Khasi*) in Bangladesh and India [32]. They are said to be 'female-centered' since *Garo* women enjoy a higher status and freedom in both the family and community. It is the privilege of the *Garo* women to choose their own partner (propose first); moreover, husbands move to live in the wife's house (matrilocal), women own all household properties and are treated as the heads of the household, all property is passed down the female line (often the younger daughter inherits the property), and children adopt matrilineal descent [32]. Although the *Garo society* maintains some features of 'matriarchy,' they lack the social order to be solely ruled by the women. The *Garo* might be the example of a female-centered and gender-equal society in rural Bangladesh.

## The Santal ethnicity

The *Santal* community is also one of the oldest and largest ethnic minority communities (about 250,000 in total) living in the northwestern part of Bangladesh. Their principal home is in *Radha* (West Bengal), the forests of adjacent Bihar, Orissa, and Chhota Nagpur in India [33]. *Santals* have their own language called Santali although most of the *Santals* in Bangladesh speak both Bangla and Santali [33]. The *Santal* society is traditionally ruled by a Panchayet system, where the village headman (manjhi) enjoys a special dignity in the society. The community is divided into twelve clans; however, marriage between a man and a woman of the same clan is prohibited. Unlike Bengalis, *Santals* maintain patrilocal and patrilineal traditions, but they are different from the patriarchal *Bengali* community. *Santals* are fond of festivities. The Spring Festival of the *Santals* provides young men and women an opportunity to exchange hearts [33]. Both women and men enjoy the freedom to choose their partner during this festival. The purpose of the festival, which takes place at the beginning of spring is to welcome and offer greetings to the beautiful blossoming of colorful flowers. It is also characterized by an overflow of dancing, singing, and music. The choice of one's partner in this ceremony is cherished by a sort of club. In the *Santal* society, there are no rules against young couples freely interacting with each other prior to marriage. Male domination is more prominent in the *Santal* society, although *Santal* women take a leading role in earning a livelihood [33]. The *Santal* women have freedom to work in public spheres. Female mobility and women's participation in economic activities are appreciated in the *Santal* community. *Santal* women have a key responsibility in their families. Inheritance rules are complex. Sons inherit most of the parental land equally, although a smaller portion may go to the daughters as dowry. *Santal* people have their own religion (called *Santana Dharma*)–though in recent years, a majority of them have converted to Christianity.

These diverse socio-cultural features among the ethnic-cultural communities indicate that it is not only the level of women's positions and gender equality, but the features of wife abuse that might be different across the ethnic communities. It is assumed that the ethnic minorities, matrilineal *Garo* as well as somehow fairly patriarchal *Santal* societies would attitudinally accept less wife abuse than that of the mainstream patriarchal Bengali community in rural Bangladesh. The aim of the current study is to assess the level of attitudinal acceptance of wife abuse among ethnic minority *Garo* and *Santal* as well as the mainstream/majority *Bengali* ethnic communities in the rural areas. By providing cross-ethnic evidence of the acceptance of wife abuse, the findings of this study may facilitate interventions to prevent widespread wife abuse in the country.

## Materials and methods

### Study design

This study used baseline data from a cluster-randomized control trial 'Community-based prevention of domestic violence among Bengali, Garo, and Santal ethnic communities in rural Bangladesh: A cross-cultural study.' The survey was conducted in February–May 2019. The focus of the main study was to examine the efficacy of a community-based intervention for the prevention of domestic violence against women. Social acceptance of wife abuse was used as one of the outcome measures of the survey related to this control trial. Although the main study is supposed to be longitudinal in nature, the current data-set represents a cross-sectional design.

### Sites under study

Fieldwork was conducted in 24 purposively selected villages–eight *Santal*, eight *Bengali*, and eight *Garo* villages, located in different parts of northwestern and northeastern Bangladesh, respectively. The *Santal* villages were selected from the northwest Upazila (sub-district) area of Bangladesh, where indigenous *Santal* communities have been living for more than 300 years. The villages were 20–30 km northwest of the Rajshahi district headquarters. The *Bengali* villages were also located in two *Upazilas* in northwest Bangladesh. Both were situated 15–20 km northeast of the Rajshahi city center. The *Garo* villages were selected from another *Upazila* located in northeast Bangladesh. The *Garo* people live in the forested area of this Upazila, where the ethnic community developed their settlements nearly 500 years ago. The *Garos* first migrated from China to the northeastern Garo-Hills area almost 4,000 years ago, and many of them later migrated to the current study villages. These villages were nearly 250km northeast of the city of Rajshahi. The primary presence of an ethnic community in an area was the main criteria for selecting the villages. There are around 68,000 villages in rural Bangladesh; therefore, it was beyond our budget and time to include a statistically representative number of villages in our study. However, we selected our study participants using a random sampling procedure.

### Study participants

The sample included currently married men and women, aged below 60 years, living in the study villages. We excluded men/women with mental/physical disabilities. As estimated during our pilot study, considering the lowest prevalence of sexual abuse in an ethnic village, 16.7%, the required minimum sample size was estimated at 1,854 using a formula ($n = z^2_{\alpha/2} p (1-p)/E^2$, where P = Proportion; $\alpha = 0.05$; therefore, $z_{\alpha/2} = 1.96$; E = p/10) [34]. To avoid non-consent, loss to follow-up, or drop out, we created a sample pool, including 10% oversamples.

In total, we approached 1,968 study participants. We used a cluster sampling procedure in order to select our study participants. At first, we randomly selected a house in a particular neighborhood community of a study village; then, we randomly included 20 nearby households of that community to form a cluster. Finally, either a married man or woman from that household was approached for face-to-face interviews. In total, 1,929 respondents (961 men and 968 married women) completed the questionnaire. The response rate was 98.02%. The non-responses were mostly related to the participants' lack of time.

## Response variable

Acceptance of wife abuse: We assessed the number of wife abuse events accepted by a respondent. We measured three aspects of the acceptance of wife abuse: acceptance of physical abuse, acceptance of emotional abuse, and acceptance of any abuse (physical or emotional). A 10-item scale was used to measure the attitudinal acceptance of wife abuse. We asked questions on context-specific reasons in rural Bangladesh, where people may justify husband's wife abusive behavior. The respondents were asked to give their opinions on whether they think a husband can abuse/mistreat his wife under a given circumstance. We provided 10 contextual scenarios related to wife's behaviors (see Table 1), where a person may endorse wife abuse, for example, 'failing to prepare meals on time.' All the proposed scenarios/events had three possible responses: 'yes–physical abuse,' 'yes–emotional abuse,' and 'none.' We provided examples of both emotional abuse (e.g., humiliation, insults, verbal rebuke, cursing, displaying anger/hatred, threatening to beat, etc.) and physical abuse (e.g., slapping, grabbing mouth, punching, kicking, beating with a fist/stick, burning, etc.) in order to rate the level of attitudinal justification of wife abuse. We constructed a score of acceptance of physical abuse by adding all the positive responses related to 'yes–physical abuse'; the score for acceptance of emotional abuse by adding the positive responses of 'yes–emotional abuse'; the score for acceptance of any abuse by adding any positive responses related to 'yes–physical abuse' and/or 'yes–emotional abuse.' The score of each of the acceptance of abuses ranged from 0 to 10, indicating the

**Table 1. Descriptives of the respondents' attitudinally accepted different wife abuses, N = 1,929.**

| Events | Any Abuse | | Emotional | | Physical | |
|---|---|---|---|---|---|---|
| | *F* | *%* | *F* | *%* | *F* | *%* |
| 1. Failing to prepare tasty meals | 262 | 13.6 | 242 | 12.5 | 20 | 1.0 |
| 2. Burning food during cooking | 275 | 14.3 | 248 | 12.9 | 27 | 1.4 |
| 3. Leaving home without getting husband's consent | 729 | 37.8 | 642 | 33.3 | 87 | 4.5 |
| 4. Failing to prepare meals on time | 319 | 16.5 | 299 | 15.5 | 20 | 1.0 |
| 5. Expecting to work or earn an income against husband's willingness | 396 | 20.5 | 327 | 17.0 | 69 | 3.6 |
| 6. Arguing with husband all the time | 932 | 48.3 | 719 | 37.3 | 213 | 11.0 |
| 7. Chatting with a man disliked by the husband | 950 | 49.2 | 721 | 37.4 | 229 | 11.9 |
| 8. Refusing sex to the husband | 216 | 11.2 | 170 | 8.8 | 46 | 2.4 |
| 9. Not wanting to have a baby when husband is interested | 603 | 31.3 | 430 | 22.3 | 173 | 9.0 |
| 10. Getting involved in an extra-marital affair | 1141 | 59.1 | 567 | 29.4 | 574 | 29.8 |
| **Scale Statistics** | | | | | | |
| Accepted wife abuse on all 10 events | 65 | 3.4 | 4 | 0.2 | 4 | 0.2 |
| Accepted wife abuse on 5/more events | 548 | 28.4 | 335 | 17.4 | 57 | 3.0 |
| Accepted wife abuse (at least 1 event) | 1381 | 71.6 | 1303 | 67.5 | 658 | 34.1 |
| | **M** | **SD** | **M** | **SD** | **M** | **SD** |
| Sum of the events accepted wife abuse (Range: 0–10) | 3.02 | 2.84 | 2.26 | 2.23 | 0.76 | 1.42 |

number of events where the respondent accepted particular wife abuse. In this study, the scale appeared to be internally consistent and reliable ($\alpha = 0.87$).

## Explanatory variables

The main explanatory variable included in the analysis was respondent's ethnicity. In the analysis, we also included some other individual level variables such as sex, age, education, and occupation as well as a few family level variables like family structure, family residence, family income, level of female mobility in the family, and level of female authority within the family.

We measured 'ethnicity' as a multinomial variable having three mutually exclusive categories: *Garo*, *Santal*, and *Bengali*. Sex had two categories: male and female. Age was classified into four ordinal levels: 16–25 years, 26–35 years, 36–45 years, and 46–60 years.

Education was measured regarding the actual years of schooling. According to the typical categorization of educational attainment in the Bangladesh context, the score was later transformed into four ordinal levels: No schooling, Primary (1–5 years of schooling), Secondary (6–10 years of schooling), and Higher (passed Secondary School Certificate examination or above).

Occupation was measured according to the subjective importance of different jobs in rural Bangladesh. The occupations were categorized into four nominal levels: Unemployed (having no formal source of income, although this may include home-making jobs); Agric Farming (earning from agriculture like producing crops, vegetables, fruits, fish-cultures, etc.); Day Laborers (earning on a daily basis by selling physical labor in agricultural, transport, or other cottage industries); and Jobs and others (earning from jobs, business, or other investments).

Monthly family income was measured on a 3-point response category, ranging from 1 = low income (earning below 9,000 Taka per month), 2 = medium income (earning from 9,000 to 14,999 Taka per month), and 3 = higher income (earning 15,000 Taka or above per month). People hardly have any fixed monthly income in rural Bangladesh; therefore, data on monthly family income were collected using a response category. Considering a poverty line family income categorization procedure and people's current socio-economic conditions in rural Bangladesh, the monthly family income was classified into these three above-mentioned ordinal levels [35].

Family structure had two categories: Nuclear (constituted with married husband and wife and/or unmarried children); and Joint (having husband/wife, unmarried/married adult children, and/or in-laws). Family residence was measured as three categories: 1 = patrilocal (wife moves to husband's parental house after marriage), 2 = matrilocal (husband moves to wife's parental house), and 3 = neutral (both separated from their parental houses and lived on their own).

Female mobility status was measured using a 6-item scale. The female respondent was asked whether she had ever visited some specific places outside of the home. On the other hand, the married male respondent was asked whether his wife had ever visited these places. We used three response categories for each of the places: 0 = no, 1 = a few times, 2 = several times. The family earned relevant female mobility-point for each of the places the woman of the family visited. For example, the questions included: 'Have you (your wife) ever been to the bazaar.' The total score ranged from 0 to 12, indicating higher female mobility with increasing scores. However, using the K-means Cluster Analysis, we further categorized the female mobility scores into three ordinal levels: Low (scored 0–3), Fair (scored 4–8), and High (scored 9–12). In this study, the scale appeared to be reliable and internally consistent ($\alpha = 0.89$).

The level of female authority within the family was measured regarding the degree of the wife's authority compared to her husband's in household decision-making [13, 36]. An 8-item

scale was used to assess who had the final say on important household decision-making issues. The context of rural Bangladesh was taken into account. For example, a sample item included 'who has the final say related to buying or selling any important household properties (e.g., homestead or faming land, house, shop, and vehicle)?' Other items were: 'who has the final say related to buying or selling crucial agricultural products (e.g., crops, livestock)?'; 'who has the final say related to developing household facilities (e.g., tube-well or water supply, furniture)?'; 'who has the final say related to inviting family friends or relatives to visit for a meal?'; 'who has the final say related to providing financial support to parents, siblings, or relatives?'; 'who has the final say related to taking sick family members to physicians or hospitals?'; and 'who has the final say related to arranging a recreational activity, like family outing or tours?' Each item was scored on a 4-point Likert-type scale, ranging from: 0 = wife has no say, 1 = wife has little say, 2 = wife has an equal say, and 3 = wife has more say. The total score ranged from 0 to 24, with higher scores indicating high female authority. However, in order to understand the relative authority structure in the family, the scores were further categorized into three levels: 3 = higher female authority (17–24), 2 = fair authority (13–16), and 1 = low authority (0–12 less). In this study, the scale showed very good internal consistency with a higher reliability, $\alpha$ = 0.93.

## Data collection

A structured questionnaire was administered in face-to-face interviews. On-site, face-to-face interviews allowed the interviewer to interact with the respondents, which ensured a clear understanding of the survey questions as well as better-quality responses [37].

The study participants were contacted in person in their homes. Four graduates in social work (two males and two females) were employed to collect the data. Male interviewers interviewed the male respondents, while female interviewers interviewed the female respondents, respectively. Due to the sensitive nature of the interviews, the interviewers were trained on ethical, safety, and technical issues related to data collection. We also trained them on how to provide support to any abused respondents who were seeking help. We emphasized the establishment of rapport with the respondents. We asked less sensitive questions first, which allowed the respondents to more easily adapt to the sensitive issues. Before having an interview, we explained the study protocols to each of the study participants. We also offered them a compensation for their time. These factors encouraged the respondents to join in the study.

## Analytical strategies

The purpose of the data analysis was to explore the factors associated with the acceptance of different types of wife abuse. The data analysis was conducted using SPSS 23.0 software [38]. Descriptive statistics of all variables were produced, which provided a profile of the study participants. We measured the acceptance of wife abuse on 10 contextual events. Thus, the frequency of accepting wife abuse (the number of events where the participant accepted wife abuse) appeared to be a count data with a good number of 'zero count' (see Table 1). Since the response variables were positively skewed count data (see Table 1), we used Poisson regression [39, 40]. Coefficients of Poisson regression with log link function, exp*(β)*, also facilitated the interpretation of regression outcomes, a proportional effect of one unit change in the explanatory variable on the response variable [39, 41]. First, bivariate analyses were conducted in order to see the ethnic differences in the other variables included in the survey questionnaire (see Table 2). Thereafter, multivariate Poisson regression was performed to explore the significant factors influencing the response variables. A line plot of the predicted mean responses was generated.

**Table 2. Demographic profile and bivariate ethnic differences in the variables, *N* = 1,929.**

| Variables | Total *F* (%) *N* = 1929 | Ethnic Distribution *F* (%) | | | $\chi^2$ | *p* |
|---|---|---|---|---|---|---|
| | | Garo $N_1$ = 640 | Santal $N_2$ = 640 | Bengali $N_3$ = 649 | | |
| **Accepted wife abuse** | | | | | | |
| Physical | 658 (34.1) | 176 (27.5) | 215 (33.6) | 267 (41.1) | 26.78 | < .001 |
| Emotional | 1303 (67.5) | 402 (62.8) | 437 (68.3) | 464 (71.5) | 11.31 | .003 |
| Any Abuse | 1381 (71.6) | 415 (64.8) | 449 (70.2) | 517 (79.7) | 35.75 | < .001 |
| **Sex** | | | | | | |
| Male | 960 (49.8) | 318 (49.7) | 319 (49.8) | 323 (49.8) | .01 | .998 |
| Female | 969 (50.2) | 322 (50.3) | 321 (50.2) | 326 (50.2) | | |
| **Age in years** | | | | | | |
| 16–25 | 301 (15.6) | 71 (11.1) | 127 (19.8) | 103 (15.9) | 130.15 | < .001 |
| 26–35 | 746 (38.7) | 220 (34.4) | 273 (42.7) | 253 (39.0) | | |
| 36–45 | 711 (36.9) | 230 (35.9) | 212 (33.1) | 269 (41.4) | | |
| 46–60 | 171 (08.9) | 119 (18.6) | 28 (04.4) | 24 (03.7) | | |
| **Education** | | | | | | |
| Higher | 385 (20.0) | 146 (22.8) | 94 (14.7) | 145 (22.3) | 53.72 | < .001 |
| Secondary | 575 (29.8) | 188 (29.4) | 166 (25.9) | 221 (34.1) | | |
| Primary | 820 (42.5) | 278 (43.4) | 304 (47.5) | 238 (36.7) | | |
| No schooling | 149 (07.7) | 28 (04.4) | 76 (11.9) | 45 (06.9) | | |
| **Occupation** | | | | | | |
| Unemployed | 636 (33.0) | 167 (26.1) | 153 (23.9) | 316 (48.7) | 420.92 | < .001 |
| Agric farming | 329 (17.1) | 154 (24.1) | 29 (04.5) | 146 (22.5) | | |
| Day laborers | 797 (41.3) | 265 (41.4) | 435 (68.0) | 97 (14.9) | | |
| Job and others | 167 (08.7) | 54 (08.4) | 23 (03.6) | 90 (13.9) | | |
| **Family Income** | | | | | | |
| 15,000/above | 478 (24.8) | 247 (38.6) | 38 (05.9) | 193 (29.7) | 256.13 | < .001 |
| 9,000–14,999 | 852 (44.2) | 296 (46.3) | 303 (47.3) | 253 (39.0) | | |
| Below 9,000 | 599 (31.1) | 97 (15.2) | 299 (46.7) | 203 (31.3) | | |
| **Family Structure** | | | | | | |
| Nuclear | 1411 (73.1) | 394 (61.6) | 511 (79.8) | 506 (78.0) | 66.01 | < .001 |
| Joint | 518 (26.9) | 246 (38.4%) | 129 (20.2%) | 143 (22.0) | | |
| **Family Residence** | | | | | | |
| Neutral | 474 (24.6) | 150 (23.4) | 162 (25.3) | 162 (25.0) | 807.63 | < .001 |
| Matrilocal | 382 (19.8) | 353 (55.2) | 21 (03.3) | 8 (01.2) | | |
| Patrilocal | 1073 (55.6) | 137 (21.4) | 457 (71.4) | 479 (73.8) | | |
| **Female Mobility** | | | | | | |
| High | 570 (29.5) | 290 (45.3) | 271 (42.3) | 9 (01.4) | 839.88 | < .001 |
| Fair | 776 (40.2) | 255 (39.8) | 337 (52.7) | 184 (28.4) | | |
| Low | 583 (30.2) | 95 (14.8) | 32 (05.0) | 456 (70.3) | | |
| **Female Authority** | | | | | | |
| Good | 274 (14.2) | 206 (32.2) | 36 (05.6) | 32 (04.9) | 513.79 | < .001 |
| Fair | 940 (48.7) | 383 (59.8) | 335 (52.3) | 222 (34.2) | | |
| Low | 715 (37.1) | 51 (08.0) | 269 (42.0) | 395 (60.9) | | |

## Ethical procedures

Ethical issues were very important in this research as we addressed people's emotions as well as their sensitive information. The study was conducted in accordance with the operational guidelines and procedures recommended by the World Health Organization for conducting

research on violence against women [42]. Ethical guidelines for public health research with human beings were also taken into account [43]. The study was conducted under the approval of a Review Committee at the Faculty of Social Sciences, University of Rajshahi, Bangladesh. All study participants were informed about the purpose and procedures of the study and their informed consent was obtained. Written consent was waived when participants were not able to read the consent form. In these cases, oral consent message was read out to them. Upon consent, the study participants were requested to suggest a suitable place and time so that the data collection process could take place in private. The participants were informed that they might find some questions uncomfortable. They were reminded repeatedly that his/her participation was completely voluntary and that she/he had no obligation to complete the interview and could drop out of the interview at any time without any further explanation. The interviewers were given intensive training about safety issues. Moreover, they were trained on the basic caring skills to help survivors of abuse so that they could, if needed, provide instant support to the abused women. To avoid any possible family discomforts, only one respondent (either male or female) from a household was selected. Anonymity and confidentiality of the interviews were maintained. We also informed the participants about our intervention. The nearest domestic violence support services were introduced to the participants who disclosed experiences of abuse and sought support.

## Results

Data revealed that the acceptance of wife abuse was high in the sample. The respondents supported any wife abuse (either physical or emotional) on an average of 3.02 events, emotional abuse 2.26 events, and physical abuse 0.76 events, regarding the scale that ranged from 0 to 10 events. Data also showed that 71.6% of the respondents accepted any wife abuse, 67.5% accepted emotional abuse, and 31.4% accepted physical abuse at least for one reason, respectively (see Table 1).

Table 2 presents the socio-demographic profile of the respondents as well as their ethnic differences. Of the sample, 33.6% were mainstream *Bengali*, 33.2% were ethnic minority *Santal*, and 33.2% were from the ethnic minority *Garo* communities. Bivariate analysis revealed that there were ethnic differences in the acceptance of different wife abuses (see Table 2). The ethnic communities also appeared to be different from each other, in terms of educational attainment, occupation, family income, family structure, family residence, female mobility, and female family authority. Table 2 shows that most of the *Santal* respondents had a lower level of education, low-status occupations like day laboring as well as a low family income. It also showed that the *Garo* society was largely matrilocal, while both the *Bengali* and *Santal* societies were predominantly patrilocal. On the other hand, female mobility was high among both the *Garo* and *Santal* communities although it was very low among the *Bengali* community. Female authority within the family unit also appeared to be quite high among the *Garo*.

The bivariate Poisson regression analysis further estimated that the *Garo* and *Santal* ethnic minority communities accepted less wife abuse than that of the majority Bengali community (see Table 3). Table 3 also represents how other explanatory variables were related to the acceptance of wife abuse. It indicates that sex, age, education, occupation, family income, family residence, women's mobility status, and female authority within the family were associated with the dependent variables. Therefore, all these variables were taken into account for the multivariate analysis.

Multivariate Poisson regression was used for exploring significant factors associated with the acceptance of wife abuse in the sample. Table 3 represents three different models explaining the acceptance of any abuse, acceptance of emotional abuse, and acceptance of physical

**Table 3. Bivariate poisson regressions for the events where wife abuses are accepted, *N* = 1,929.**

| Variables | Any Abuse | | | | Emotional | | | | Physical | | |
|---|---|---|---|---|---|---|---|---|---|---|---|
| | *B* | *Exp(B)* | $\chi^2$ | | *B* | *Exp(B)* | $\chi^2$ | | *B* | *Exp(B)* | $\chi^2$ |
| **Ethnicity** | | | | | | | | | | | |
| Garo | -0.32 | 0.73 | 95.36*** | | -0.18 | 0.83 | 24.75*** | | -0.72 | 0.49 | 114.10*** |
| Santal | -0.19 | 0.83 | 37.86*** | | -0.12 | 0.88 | 10.63*** | | -0.39 | 0.68 | 41.49*** |
| Bengali | | 1 | | | | 1 | | | | 1 | |
| **Sex** | | | | | | | | | | | |
| Male | -0.89 | 0.41 | 945.14*** | | -0.33 | 0.47 | 513.29*** | | -1.37 | 0.25 | 440.04*** |
| Female | | 1 | | | | 1 | | | | 1 | |
| **Age in years** | | | | | | | | | | | |
| 16–25 | 0.46 | 1.58 | 59.55*** | | 0.44 | 1.55 | 40.45*** | | 0.49 | 1.63 | 19.21*** |
| 26–35 | 0.32 | 1.38 | 34.60*** | | 0.36 | 1.44 | 32.46*** | | 0.20 | 1.22 | 3.53^ |
| 36–45 | 0.21 | 1.23 | 13.70*** | | 0.25 | 1.29 | 15.33*** | | 0.06 | 1.06 | 0.35 |
| 46–60 | | 1 | | | | 1 | | | | 1 | |
| **Education** | | | | | | | | | | | |
| Higher | -0.46 | 0.63 | 72.36*** | | -0.24 | 0.79 | 13.70*** | | -0.97 | 0.38 | 97.84*** |
| Secondary | -0.19 | 0.83 | 15.67*** | | -0.02 | 0.98 | 0.07 | | -0.58 | 0.56 | 48.59*** |
| Primary | -0.19 | 0.82 | 17.22*** | | 0.01 | 1.00 | 0.01 | | -0.66 | 0.52 | 67.20*** |
| No schooling | | 1 | | | | 1 | | | | 1 | |
| **Occupation** | | | | | | | | | | | |
| Unemployed | 0.82 | 2.28 | 188.38*** | | 0.67 | 1.95 | 188.38*** | | 1.34 | 3.82 | 90.65*** |
| Agric farming | 0.24 | 1.28 | 13.17*** | | 0.19 | 1.21 | 13.17* | | 0.46 | 1.59 | 8.77*** |
| Day laborers | 0.33 | 1.39 | 29.40** | | 0.28 | 1.32 | 29.40*** | | 0.56 | 1.75 | 14.98*** |
| Job and others | | 1 | | | | 1 | | | | 1 | |
| **Family Income** | | | | | | | | | | | |
| 15,000/above | 0.01 | 1.00 | 0.01 | | 0.02 | 1.02 | 0.30 | | -0.05 | 0.95 | 0.56 |
| 9,000–14,999 | -0.09 | 0.91 | 8.80** | | -0.05 | 0.95 | 2.04 | | -0.21 | 0.81 | 11.95*** |
| Below 9,000 | | 1 | | | | 1 | | | | 1 | |
| **Family Structure** | | | | | | | | | | | |
| Nuclear | 0.04 | 1.04 | 2.07 | | 0.06 | 1.06 | 3.17^ | | 0.06 | 0.99 | 0.04 |
| Joint | | 1 | | | | 1 | | | | 1 | |
| **Family Residence** | | | | | | | | | | | |
| Neutral | 0.26 | 1.30 | 78.30*** | | 0.27 | 1.31 | 62.79*** | | 0.23 | 1.26 | 15.83*** |
| Matrilocal | -0.15 | 0.86 | 16.20*** | | -0.11 | 0.90 | 6.34* | | -0.28 | 0.75 | 13.55*** |
| Patrilocal | | 1 | | | | 1 | | | | 1 | |
| **Female Mobility** | | | | | | | | | | | |
| High | -0.50 | 0.61 | 188.01*** | | -0.42 | 0.65 | 100.94*** | | -0.74 | 0.48 | 97.56*** |
| Fair | -0.01 | 0.99 | 0.13 | | 0.05 | 1.05 | 1.88 | | -0.17 | 0.84 | 9.14*** |
| Low | | 1 | | | | 1 | | | | 1 | |
| **Female Authority** | | | | | | | | | | | |
| Good | -0.51 | 0.60 | 136.78*** | | -0.50 | 0.61 | 97.63*** | | -0.55 | 0.58 | 39.29*** |
| Fair | -0.43 | 0.65 | 240.01*** | | -0.41 | 0.66 | 161.85*** | | -0.50 | 0.61 | 79.94*** |
| Low | | 1 | | | | 1 | | | | 1 | |

^$p < .10$

*$p < .05$

**$p < .01$

***$p < .001$, *B* = unstandardized regression coefficient, $\chi^2$ = Walid Chi-square value

abuse. The table indicates that both the *Garo* and *Santal* communities accepted wife abusive events significantly less than the Bengali community. Holding other explanatory factors constant, the acceptance of any wife abuse was reflected as being 16% less among the ethnic-minority *Garo* and 14% less among the ethnic-minority *Santal* communities than that of the mainstream *Bengali* community (see Table 4). Nonetheless, the acceptance of emotional wife

**Table 4. Multivariate poisson regressions for the events where wife abuses are accepted, *N* = 1929.**

| Variables | Any Abuse | | | Emotional | | | Physical | | |
|---|---|---|---|---|---|---|---|---|---|
| | *B* | *SE* | *Exp(B)* | *B* | *SE* | *Exp(B)* | *B* | *SE* | *Exp(B)* |
| **Ethnicity** | | | | | | | | | |
| Garo | -0.18*** | 0.05 | 0.84 | -0.03 | 0.05 | 1.03 | -0.82*** | 0.10 | 0.44 |
| Santal | -0.16*** | 0.04 | 0.86 | -0.06 | 0.05 | 0.94 | -0.40*** | 0.08 | 0.67 |
| Bengali | | | 1 | | | 1 | | | 1 |
| **Sex** | | | | | | | | | |
| Male | -1.01*** | 0.04 | 0.37 | -0.82*** | 0.05 | 0.44 | -1.66*** | 0.09 | 0.19 |
| Female | | | 1 | | | 1 | | | 1 |
| **Age in years** | | | | | | | | | |
| 16–25 | -0.25*** | 0.07 | 0.78 | -0.13 | 0.08 | 0.88 | -0.66*** | 0.13 | 0.52 |
| 26–35 | -0.19** | 0.06 | 0.83 | -0.04 | 0.07 | 0.96 | -0.69*** | 0.12 | 0.50 |
| 36–45 | -0.08 | 0.06 | 0.92 | 0.05 | 0.07 | 1.05 | -0.53*** | 0.11 | 0.59 |
| 46–60 | | | 1 | | | 1 | | | 1 |
| **Education** | | | | | | | | | |
| Higher | -0.25*** | 0.06 | 0.77 | -0.07 | 0.07 | 0.93 | -0.66*** | 0.11 | 0.51 |
| Secondary | 0.01 | 0.05 | 1.01 | 0.15* | 0.06 | 1.16 | -0.28** | 0.09 | 0.75 |
| Primary | 0.04 | 0.05 | 1.03 | 0.19*** | 0.06 | 1.21 | -0.28** | 0.08 | 0.75 |
| No schooling | | | 1 | | | 1 | | | 1 |
| **Occupation** | | | | | | | | | |
| Unemployed | 0.01 | 0.07 | 1.00 | -0.02 | 0.08 | 0.98 | 0.05 | 0.15 | 1.05 |
| Agric farming | 0.19** | 0.07 | 1.21 | 0.15* | 0.08 | 1.16 | 0.37* | 0.16 | 1.44 |
| Day laborers | 0.14* | 0.06 | 1.15 | 0.10 | 0.07 | 1.11 | 0.29 | 0.15 | 1.33 |
| Job and others | | | 1 | | | 1 | | | 1 |
| **Family Income** | | | | | | | | | |
| 15,000/above | -0.08 | 0.04 | 0.92 | -0.05 | 0.04 | 0.95 | -0.17* | 0.07 | 0.84 |
| 9,000–14,999 | -0.01 | 0.03 | 1.00 | 0.02 | 0.04 | 1.02 | -0.06 | 0.06 | 0.94 |
| Below 9,000 | | | 1 | | | 1 | | | 1 |
| **Family Structure** | | | | | | | | | |
| Nuclear | -0.01 | 0.03 | 1.00 | 0.03 | 0.04 | 1.03 | -0.07 | 0.06 | 0.93 |
| Joint | | | 1 | | | 1 | | | 1 |
| **Family Residence** | | | | | | | | | |
| Neutral | 0.08* | 0.03 | 1.09 | 0.09* | 0.04 | 1.09 | 0.06 | 0.06 | 1.06 |
| Matrilocal | 0.03 | 0.05 | 1.03 | 0.02 | 0.05 | 0.98 | 0.18 | 0.10 | 1.20 |
| Patrilocal | | | 1 | | | 1 | | | 1 |
| **Female Mobility** | | | | | | | | | |
| High | -0.23*** | 0.05 | 0.80 | -0.22*** | 0.05 | 0.80 | -0.26** | 0.09 | 0.77 |
| Fair | -0.12*** | 0.04 | 0.89 | -0.07 | 0.04 | 0.93 | -0.28*** | 0.07 | 0.76 |
| Low | | | 1 | | | 1 | | | 1 |
| **Female Authority** | | | | | | | | | |
| Good | -0.23*** | 0.05 | 0.79 | -0.33*** | 0.06 | 0.72 | -0.01 | 0.09 | 0.99 |
| Fair | -0.21*** | 0.03 | 0.81 | -0.26*** | 0.04 | 0.77 | -0.07 | 0.06 | 0.93 |

*(Continued)*

**Table 4.** (Continued)

| Variables | Any Abuse | | | Emotional | | | Physical | | |
|---|---|---|---|---|---|---|---|---|---|
| | *B* | *SE* | *Exp(B)* | *B* | *SE* | *Exp(B)* | *B* | *SE* | *Exp(B)* |
| Low | | | 1 | | | 1 | | | 1 |
| **Model Summary** | | | | | | | | | |
| Log Likelihood | -4425.50 | | | -3939.43 | | | -2313.41 | | |
| AIC | 8895.01 | | | 7922.86 | | | 4670.82 | | |

*$p < .05$

**$p < .01$

***$p < .001$, *B* = unstandardized regression coefficient, *SE* = Standard Error

abuse did not appear to be significantly different among the minority communities from the mainstream community. However, the acceptance of physical abuse of the wife appeared to be 56% lower among the ethnic *Garo* and 33% lower among the ethnic *Santal* than the *Bengali* community (see Table 4).

Besides ethnicity, the study showed that the rate of acceptance of any wife abuse was 63% less, emotional abuse 56% less, and physical abuse 81% less among the male respondents than the female respondents (see Table 4). It also appeared that relatively younger generations were less accepting of any abuse and physical abuse than the aged respondents. People with higher education also appeared to be less accepting of any abuse 23% and physical abuse 49% than the respondents who had no schooling. Agric farmers were also more likely to accept all three types of abuse than the respondents who engaged in jobs/business (see Table 4). Regardless of the ethnicity and other factors, the study also indicated that the acceptance of any abuse was 20% less, emotional abuse 20% less, and physical abuse 23% less among the families where female mobility was high compared to families with low female mobility. The acceptance of any abuse and emotional abuse was 21% less and 28% less, respectively, among the families with higher female authority than with low female authority (see Table 4).

The Likelihood Ratio Chi-square statistics for the three different models as presented in Table 4 further demonstrate that the independent variables included in the analysis produced statistically significant overall models for explaining the acceptance of wife abuse (Any abuse $\chi^2 = 1395.00$, df = 21, $p < .001$; Emotional Abuse $\chi^2 = 799.39$, df = 21, $p < .001$; and Physical Abuse $\chi^2 = 813.95$, df = 21, $p < .001$). The test of model effects also revealed that the ethnicity produced a significant amount of variance for the acceptance of any wife abuse ($\chi^2 = 18.27$, df = 2, $p < .001$), as well as for the acceptance of physical abuse ($\chi^2 = 68.01$, df = 2, $p < .001$). The predicted means plot of the acceptance of different wife abuse additionally showed that both the ethnic minority *Garo* and *Santal* were less accepting of wife abuse events than the *Bengali* community (see Fig 1).

## Discussion

This study examines the ethnic differences in the acceptance of different wife abuse events among the ethnic-minority *Garo* and *Santal* as well as the mainstream *Bengali* communities in rural Bangladesh. From the findings, it appears that the attitudinal acceptance of wife abuse is very high in the country. Data also show that the acceptance of wife's emotional abuse is higher than the acceptance of the wife's physical wife. The study reveals that the acceptance of all types of wife abuse is significantly lower among the ethnic-minority *Garo* and *Santal* communities than the mainstream *Bengali* community. Notably, the matrilineal *Garo* ethnic

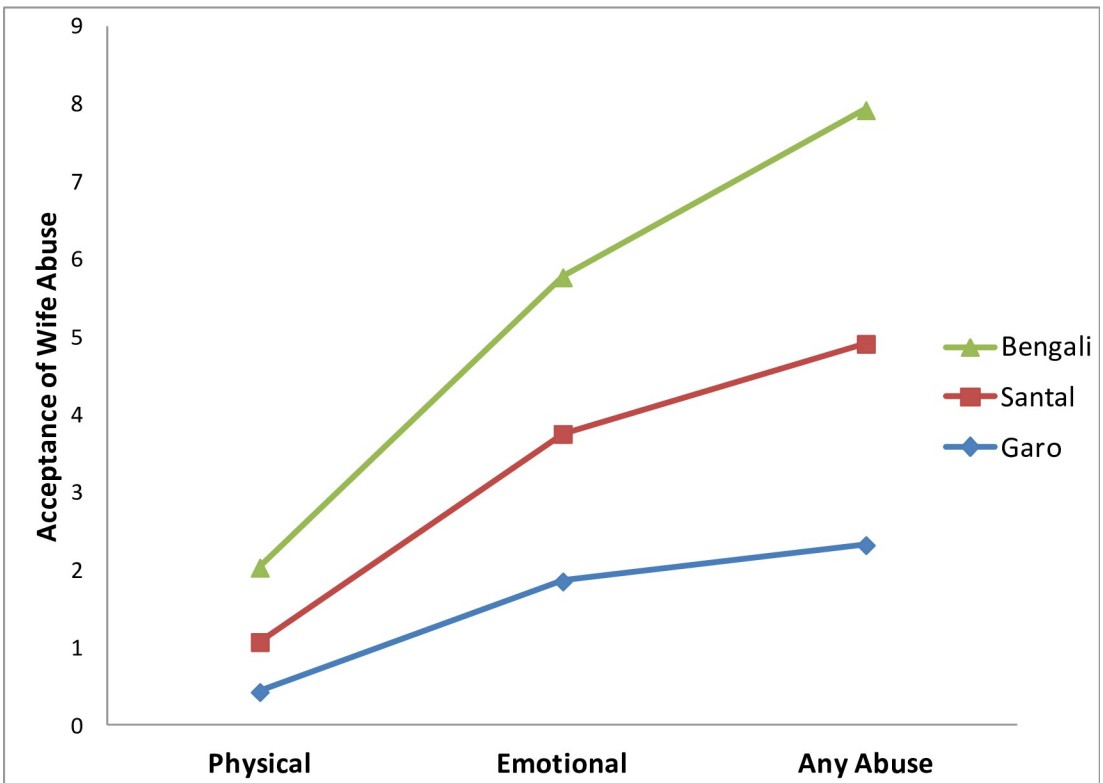

**Fig 1. Predicted means plot of the acceptance of wife abuse events by ethnicity.**

community has the lowest rates of acceptance of all wife abuses among the three communities. Our study further reveals that individual's higher level of education as well as their family's higher level of income, higher level of mobility of the women in his/her family, and the higher female authority within his/her family were inversely correlated with the acceptance of wife abuse.

The findings of our study are consistent with a body of previous studies that show a very high acceptance of wife's physical abuse in rural Bangladesh [4, 12, 24, 44–46]. The previous studies mostly estimated the rates of the acceptance of wife beating [4, 24, 28, 45], whereas our study includes the estimates related to the wife's emotional and physical abuse. This is why our findings might apparently estimate a higher rate of acceptance of wife abuse than previous studies conducted in the country [3, 47]. Both the studies of Jesmin [47] and Rani [3] used large-scale data sets managed by reputed health organizations, indicating that 32% and 36.6%, respectively, of Bangladeshi people accept wife beating. Our study, on the other hand, categorically estimates that 34.1% of the respondents accepted the wife's physical abuse, while 67.5% accepted emotional abuse and 71.6% accepted either physical or emotional abuse. These differences are probably because our study incorporates more comprehensive (e.g., both physical and emotional abuses) and sensitive issues (e.g., if the woman is involved in an extra-marital affair) [48]. Furthermore, establishing rapport with the interviewees, intensive training and supervision of the interviewers, and thorough observation of ethical guidelines might have produced reliable data in this study, which is sometimes difficult to maintain in large-scale studies [49].

While previous studies conducted in Bangladesh incorporated data from married women only [4, 24, 28, 45], our study includes the estimates of both men and women, Surprisingly, it

revealed that married women are more likely to be accepting of wife abuse than their male counterparts. These findings are also consistent with the findings of other studies where it shows that women in Bangladesh as well as in many lower-income countries uphold more conservative attitudes toward women and wife abuse than the men in their societies [26, 44, 50].

Previous studies hardly assessed the acceptance of wife abuse among the ethnic minority communities in rural Bangladesh. In that sense, the current study contributes to the literature with further knowledge. It shows that both the ethnic minority communities in our study sites differ from the mainstream *Bengali* community, not only in terms of their attitudes toward wife abuse but related to other social features like status of women in their societies, e.g., female mobility and female family authority. Although the magnitude and intensity of the acceptance of wife abuse are higher among the mainstream *Bengali* community, the findings reveal that ethnic minorities also accept wife abuse, particularly on sensitive issues. These findings also challenged a myth that the matrilineal ethnic community might not accept wife abuse [51]. Nonetheless, our field experiences indicate that the *Garo* community hardly accept wife's physical abuse, even on crucial issues like extra-marital affair; rather, they are more used to endorsing humiliation/verbal rebuke.

Broadly, the findings of our study also support the propositions of the patriarchy theory of domestic violence, where it is argued that wife abuse is a symptom of patriarchal social order/structure endorsed by patriarchal gender norms [20, 52]. We believed that the gender regime of a society may have a great influence on their people's attitudes toward wife abuse. Our study also reveals that the acceptance of wife abuse is varied according to the gendered social structure of the communities. The acceptance of wife abuse is significantly lower among ethnic minority *Garo* and *Santal* communities than among the mainstream *Bengali* community. The study also revealed that the matrilineal *Garo* community has the lowest level of acceptance of wife abuse among the three ethnic communities. This is probably because of their 'female-centered' social structure, where *Garo* women are not only highly valued but the level of gender equality appeared quite high, e.g., women's increased access to resources, their free mobility in public spheres as well as their greater ability to make household decisions. In the patriarchal *Santal* community, women are also expected to earn an income and are more free to move in public spheres. However, in the *Bengali* community, women are supposed to be confined within the home-boundary. The patriarchal social structure and their norms not only restrict *Bengali* women's economic independence and social status but force them to be controlled by their husbands.

## Limitations and future research directions

There are some limitations that should be noted in order to understand the findings of the current study. Although the study incorporates an adequate number of observations from three ethnic communities, the findings should probably be limited to that particular area. To make generalizations, it is necessary to have more representative samples, including other matrilocal (e.g., Khasi) as well as ethnic minority communities located in the southeastern part of the country.

The current study conceptualized the acceptance of wife abuse as one-dimensional, which might also preclude the possibility of multiple, complex, and/or combined factors related to the response variable. The acceptance of wife abuse might be different for the scenarios related to 'disobeying family obligations' compared to reasons such as 'challenging male authority.'

## Conclusions

The attitudinal acceptance of wife abuse is very high in rural Bangladesh. These findings may indicate one of the reasons as to why wife abuse is so widespread in the country. However, the

ethnic minorities appear to be less accepting of wife abuse than the mainstream community. Since most of the people justify wife abuse under certain circumstances, it is crucial to address the issue from a normative/ideological perspective. The current study reveals that a gender equal social order, as reflected in a woman's increased socio-economic status within ethnic minority communities, may reduce the social acceptance of wife abuse. These findings may have significant implications for the prevention of wife abuse in rural Bangladesh. We argue that gender equality is the key to preventing widespread wife abuse in a patriarchal society.

## Policy recommendations

We observe that societal level factors such as the gender regime of a society has a great influence on the acceptance of wife abuse in that society. It indicated that women would have more safety within the marriage if the society can uphold a 'gender-sensible' socio-cultural mechanism, where women may have the freedom to move in public spheres, equitable access to economic resources, and ability to make/implement important decisions. We argue that, beyond individual and family level initiatives, a comprehensive transformation of the patriarchal social order is imperative to reduce the widespread wife abuse in rural Bangladesh. There should be societal level initiatives for changing people's attitudes toward wife abuse. There should also be appropriate initiatives to allow women to freely move in public spheres. It is the responsibility of a society to ensure that women have free mobility, beyond their home boundaries. In order to address the societal level gender inequalities, there should be laws and regulations so that all women can have equitable access to socio-economic resources, including the inheritance of parental property. Individual women should also have access to education so that they can realize the worth of equality.

## Supporting information

**S1 File.**
(DOCX)

**S2 File.**
(XLSX)

**S3 File.**
(SAV)

## Author Contributions

**Conceptualization:** Rabiul Karim, Katarina Swahnberg.

**Data curation:** Rabiul Karim, Hafijur Rahman, Suchona Rahman.

**Formal analysis:** Rabiul Karim.

**Funding acquisition:** Rabiul Karim, Sadequl Arefin, Katarina Swahnberg.

**Investigation:** Rabiul Karim, Sadequl Arefin, Hafijur Rahman, Suchona Rahman.

**Methodology:** Rabiul Karim, Katarina Swahnberg.

**Project administration:** Rabiul Karim, Tanzima Zohra Habib, Sadequl Arefin, Katarina Swahnberg.

**Resources:** Rabiul Karim.

**Software:** Rabiul Karim.

**Supervision:** Katarina Swahnberg.

**Validation:** Rabiul Karim, Hafijur Rahman, Suchona Rahman, Katarina Swahnberg.

**Writing – original draft:** Rabiul Karim.

**Writing – review & editing:** Tanzima Zohra Habib, Katarina Swahnberg.

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
