## [Decision Letter · Decision Letter 0]

29 Apr 2020

PONE-D-20-07653

Is Wife Abuse Accepted in Matrilocal Communities? A Cross-ethnic Study in Rural Bangladesh

PLOS ONE

Dear Dr. Karim,

Thank you for submitting your manuscript to PLOS ONE. After careful consideration, we feel that it has merit but does not fully meet PLOS ONE’s publication criteria as it currently stands. Therefore, we invite you to submit a revised version of the manuscript that addresses the points raised during the review process.

Reviewer one in particular raises serious concerns about the validity of your approach. It is important to report the results in a statistically sound manner and this means with standard errors and confidence intervals for the marginals. I need you to give a robust response to the criticism about your not using the demographic norms established in the existing literature otherwise it is likely that the paper will be rejected.

We would appreciate receiving your revised manuscript by Jun 13 2020 11:59PM. To enhance the reproducibility of your results, we recommend that if applicable you deposit your laboratory protocols in protocols.io, where a protocol can be assigned its own identifier (DOI) such that it can be cited independently in the future. For instructions see: http://journals.plos.org/plosone/s/submission-guidelines#loc-laboratory-protocols

We look forward to receiving your revised manuscript.

Kind regards,

Andrew R. Dalby, PhD

Academic Editor

PLOS ONE

Journal Requirements:

2. For studies involving humans categorized by race/ethnicity, age, disease/disabilities, religion, sex/gender, sexual orientation, or other socially constructed groupings, authors should: 1) Explicitly describe their methods of categorizing human populations, 2) Define categories in as much detail as the study protocol allows, 3) Justify their choices of definitions and categories, 4) Explain whether (and if so, how) they controlled for confounding variables such as socioeconomic status, nutrition, environmental exposures, or similar factors in their analysis.

3. Please change your reference to "p=0.000" to "p<0.001" or as similarly appropriate, as p values cannot equal zero.

Additional Editor Comments (if provided):

Reviewers' comments:

Reviewer's Responses to Questions

**Comments to the Author**

1. Is the manuscript technically sound, and do the data support the conclusions?

Reviewer #1: No

Reviewer #2: Partly

2. Has the statistical analysis been performed appropriately and rigorously? 

Reviewer #1: No

Reviewer #2: Yes

3. Have the authors made all data underlying the findings in their manuscript fully available?

Reviewer #1: No

Reviewer #2: Yes

4. Is the manuscript presented in an intelligible fashion and written in standard English?

Reviewer #1: No

Reviewer #2: No

5. Review Comments to the Author

Reviewer #1: This paper uses data from a purposively selected six villages located in northern Bangladesh to study differences in (i) ethnic characteristics of wives and (ii) attitudinal acceptance of wives abuse, more generally the domestic violence against themselves, by keeping both the research question and econometric specifications unstated. The authors collectively decided not to connect to any of the strands in the literature, -on the effects of historical social norms on present day economic outcomes- especially the glaring one (Alesina, Alberto, Benedetta Brioschi, and Eliana La Ferrara. “Violence Against Women: A Cross-cultural Analysis for Africa”). Without reckoning the detailed ethnography available in Murdock's (1967) Ethnographic Atlas, where more than 95 social norms are coded for both the communities the authors purposively chose to study (The Garos and The Santhals). It is widely recognized in the literature that the post-marital residence norms are not practiced in isolation, but concurrently with several other norms. In the limitation about the external validity of the seemingly non-causal (perhaps spurious) correlations, even without reporting the marginal effects and their corresponding standard errors is not a tenable academic practice, the authors admittedly accept the limited sample size. This could be avoided using several rounds of nationally-representative samples from Demographic and Health Survey data for Bangladesh, which covers the attitudinal measures towards wives acceptance of abuse. Much greater variations in ethnicity can be achieved if geographic data is exploited and obviously the relevant the literature on how to identify ethnicity with the spatial information and census data needs to be acknowledged if the methods are adopted.

Reviewer #2: The study is about a very interesting topic in an understudied context. The article provides significant contribution on gender based violence. The text is clear and understandable enough. Before publication, the following should be considered:

I. Abstract: Needs to strengthened the on key findings of the research and no clearly policy relevance recommendations

II. Introduction: The introduction has well written about the contexts. However, lacks some depth about the different literatures in other countries and the extent of the situation.

III. Methodology: How the sample 331 calculated? What is the scientific ground in selecting the sample size? How about the tools for measurement- do you tested the tool or are using the standard measurement tool ? Ethics section not compreleted “The study was conducted under the approval of Faculty Review Committee at …………………….”

IV. Results and discussion

I would suggest you to avoid repetitions, either put in text, tables or figures. Some results are also repeated in discussion.

V. Conclusion and recommendations: Authors did not show the relevance and policy implication of this study. I could not see a concrete recommendation for decision makers and planners.

6. PLOS authors have the option to publish the peer review history of their article (what does this mean?). If published, this will include your full peer review and any attached files.

Reviewer #1: Yes: Sumantra Pal

Reviewer #2: No

---

## [Author Response · Author response to Decision Letter 0]

16 Jun 2020

We have addressed all the comments and suggestions as follows (also in the attached file response to the reviewers):

Reviewer #1: 

1) We reviewed the above-mentioned literatures as well as other studies including another multi-country study based on the DHS dataset from 49 countries. Thanks for suggesting other useful literatures. We may now say that we have connected to the strands in the literature and further justified the importance of our current study (see lines: 87-108)

2) We changed our main focus from post-marital residence norms to the gender norms of the three ethnic communities. The title of the paper is changed (line. 4). We further elaborated how these three ethnic communities are socio-culturally different from each other. The discussion includes their post-marital residence norms, line of descent adopted by the children, inheritance of property to the children by gender, status of female mobility within the family (by veiled seclusion, many women have restricted mobility in to public spheres), women’s participation in economic activities, and women’s status within the family as to their ability to make family decisions. (See lines: 129-190)

3) We tried to avoid spurious relationship through a multivariate analysis, where we controlled the number of possible confounding factors such as respondents’ sex, age, education, occupation, family income, family structure, post-marital family residence, level of female mobility in the family, and level of women’s authority in the family. All these variables appeared to be associated with the independent variables in bivariate analysis. (See lines: 379-400, 411-417).

4) We used SPSS for the Poisson regression in order to estimate the difference in acceptance of wife abuse (AWA) among three ethnic communities. We reported the unstandardized regression coefficients and their corresponding standard errors. Although Average Marginal effects (AME) are simpler to interpret and understand and also are not affected by extreme values, unfortunately SPSS may not calculate AME. However, in Poisson regression, SPSS also produces similar standardized estimates exp(β), which also facilitate the interpretation of regression outcomes, a proportional (often expressed in percentages) effect of one unit change in the explanatory variable on the response variable. 

5) Now we used data from our baseline survey. This includes 1,929 samples from 24 study villages. The findings of this dataset are consistent with the previous pilot study data.

6) The DHS survey only includes the women of reproductive age. However, our data include both male and female respondents. Based on our knowledge, we have not seen any previous DHS data-based studies that have controlled/reported on the ethnic variations of respondents in the Bangladeshi context. Ethnic minorities often reside in very remote villages (not always easy to reach) of Bangladesh. Without any special attention, any representative sample may exclude these people. More importantly, we included both the acceptance of physical abuse and the acceptance of emotional abuse, while DHS surveys only include the acceptance of wife beating. The DHS questionnaire uses 6 items, whereas we used 10 items. The DHS survey excluded very important events like where a person accepts wife abuse because of wife’s suspected extra-marital affairs. Previous studies indicate that the most common contextual reason for wife abuse in Bangladesh is related to women’s challenges of male authority, e.g., if wife argues with the husband, and on the suspected extra-marital relations. Therefore, we believe that our data are more comprehensive and relates to our study aim. 

Reviewer #2: 

1) The abstract is revised accordingly. We added the policy recommendations. Thank you.

2) The literature review is now enriched with the examples of other countries. (please see lines: 55-58, 87-108).

3) We further revised our findings based on a new larger sample. We used data from the 1,929 sample (the baseline data from 24 villages). The previous version of the analysis was based on our pilot study, where we interviewed 383 respondents in 6 study villages. We revised as well as validated our study instruments during the pilot study. However, since we now have the baseline data from a larger sample on the same variables, we revised our analysis based on this new/large data set. The technique of minimum sample size determination as well as how we selected the samples are now elaborated (see lines: 228-240). We also further discussed the psychometric properties of our study measures including the sample of items, their internal consistency statistics, supporting literatures, calculation of scores, and their further categorization, etc.: acceptance of wife abuse (lines: 242-261), female mobility (lines: 294-303), and female authority (lines: 304-321).

4) We have further added to the ethical section now. Missing information has been included (lines: 351-371).

5) We tried to avoid repetitions. But some explanation of the table/figure data might be needed in the text. We tried to make them as few as possible. Thank you. 

6) Discussions are further tightened (highlighted red) and a separate section on policy recommendation is added. (Lines: 532-546). Thank you so much.

---

## [Decision Letter · Decision Letter 1]

14 Jul 2020

Differences in the Acceptance of Wife Abuse among ethnic minority Garo and Santal and mainstream Bengali Communities in Rural Bangladesh

PONE-D-20-07653R1

Dear Dr. Karim,

We’re pleased to inform you that your manuscript has been judged scientifically suitable for publication and will be formally accepted for publication once it meets all outstanding technical requirements.

Kind regards,

Andrew R. Dalby, PhD

Academic Editor

PLOS ONE

Additional Editor Comments (optional):

Reviewers' comments:

Reviewer's Responses to Questions

**Comments to the Author**

1. If the authors have adequately addressed your comments raised in a previous round of review and you feel that this manuscript is now acceptable for publication, you may indicate that here to bypass the “Comments to the Author” section, enter your conflict of interest statement in the “Confidential to Editor” section, and submit your "Accept" recommendation.

Reviewer #2: All comments have been addressed

2. Is the manuscript technically sound, and do the data support the conclusions?

Reviewer #2: Yes

3. Has the statistical analysis been performed appropriately and rigorously? 

Reviewer #2: Yes

4. Have the authors made all data underlying the findings in their manuscript fully available?

Reviewer #2: (No Response)

5. Is the manuscript presented in an intelligible fashion and written in standard English?

Reviewer #2: Yes

6. Review Comments to the Author

Reviewer #2: The authors has gone through all the feedbacks and substantial improvement was made through out the paper. The authors has made improvement from adjusting the background, methodology including the methodology section, and all other sections. Therefore, the manuscript can be accepted.

7. PLOS authors have the option to publish the peer review history of their article (what does this mean?). If published, this will include your full peer review and any attached files.

Reviewer #2: **Yes: **Muluken Dessalegn

---

## [Editor Report · Acceptance letter]

15 Jul 2020

PONE-D-20-07653R1 

Differences in the Acceptance of Wife Abuse among ethnic minority Garo and Santal and mainstream Bengali Communities in Rural Bangladesh 

Dear Dr. Karim:

I'm pleased to inform you that your manuscript has been deemed suitable for publication in PLOS ONE. Congratulations! Your manuscript is now with our production department. 

Kind regards, 

on behalf of

Dr. Andrew R. Dalby 

Academic Editor

PLOS ONE